# Correlation between choroidal structure and smoking in eyes with central serous chorioretinopathy

Kazuyoshi Okawa[1], Tatsuya Inoue[1]*, Ryo Asaoka[2,3,4], Keiko Azuma[2], Ryo Obata[2], Rei Arasaki[1], Shouko Ikeda[1], Arisa Ito[1], Maiko Maruyama-Inoue[1], Yasuo Yanagi[1], Kazuaki Kadonosono[1]

1 Department of Ophthalmology and Micro-Technology, Yokohama City University, Kanagawa, Japan,
2 Department of Ophthalmology, The University of Tokyo, Graduate School of Medicine, Tokyo, Japan,
3 Department of Ophthalmology, Seirei Hamamatsu General Hospital, Shizuoka, Japan, 4 Seirei Christopher University, Shizuoka, Japan

* inouet-tky@umin.ac.jp

## Abstract

### Purpose

A smoking habit can cause various health problems encompassing retinal diseases including central serous chorioretinopathy (CSC). The aim of the current study was to investigate the effect of smoking on the choroidal structure in patients with CSC.

### Methods

The choroidal vascular index (CVI) was calculated using the binarized OCT images. Baseline parameters (age, refractive error [SE], subfoveal choroidal thickness [SFCT] and CVI) were compared between smokers and non-smokers using Wilcoxon rank sum test. Moreover, the associations between SFCT and the baseline parameters were analyzed using a multivariate linear regression followed by the AICc model selection.

### Results

Among 75 CSC patients, 45 patients were smokers and 30 patients were non-smokers. No significant differences in age and SE were seen between the smoking group and the non-smoking group. A significant difference in the SFCT was seen between two groups (382.0 ± 68.2 μm in the smoking group vs. 339.3 ± 52.3 μm in the non-smoking group, $p = 0.0038$), while no significant difference was observed in the CVI ($p = 0.32$). The optimal model for SFCT included the variables of age, SE and past history of smoking among the baseline parameters. Additionally, increased pack years was associated with increased SFCT.

### Conclusion

Cigarette smoking was associated with an increased SFCT in patients with CSC. Thicker choroid in smoking CSC patients may be an important modulator of the disease.

**Data Availability Statement:** All relevant data are within the manuscript and its Supporting Information files.

**Funding:** The authors received no specific funding for this work.

**Competing interests:** The authors have declared that no competing interests exist.

## Introduction

Central serous chorioretinopathy (CSC) is characterized by serous retinal detachment accompanied by the dysfunction of the retinal pigment epithelium (RPE). Recent studies have clarified the change in the choroid. First, CSC is generally characterized by choroidal vascular hyperpermeability (CVH) on indocyanine green angiography [1–3]. Additionally, enhanced depth imaging (EDI-OCT) [4], which allows for the visualization of the choroidal structure, clarified the ratio of the luminal area to the total choroidal area, known as the choroidal vascular index (CVI) to be increased in CSC. As such, functional and anatomical abnormalities of the choroid (so-called "pachychoroid") are considered to be associated with CSC [5,6]. Specifically, increased osmotic pressure, presumably due to an increased area of the choroidal vessels, is thought to be associated with choroidal hyperpermeability, causing RPE detachment, leading to the accumulation of fluid in the subretinal space. Interestingly, eyes with acute CSC had a higher CVI than eyes without CSC or eyes with resolved CSC [7], suggesting that CVI is dynamically associated with CSC, which may change during the course of disease depending on the activity.

A smoking habit is a modifiable risk factor that is associated with retinal diseases including CSC [8–11]. Interestingly, recent detailed examinations of choroidal structural changes in healthy smokers has suggested that the CVI was significantly smaller in smokers than in non-smokers whereas the foveal retinal thickness (FRT) and the subfoveal choroidal thickness (SFCT) were unchanged [12]. Although the pathological significance of such differences in healthy subjects remains unclear, it raises a possibility that there may be some differences in the choroidal pathologies of CSC patients between smokers and non-smokers. Interestingly, smoking was reportedly associated with poor visual outcome in CSC patients [13]. Since the choroidal flow, as well as choroidal thickness, seems to have abnormal regulation in CSC patients, a smoking habit in CSC patients may be related with not only CVI, but also choroidal thickness. The hypothesis prompted us to investigate the effect of a smoking habit on the choroidal structure, including the choroidal thickness and CVI, in patients with CSC.

## Methods

The present study was a cross-sectional study conducted at a single center. The medical records of patients with CSC were retrospectively reviewed. This study was approved by the Ethics Committee of the Yokohama City University Medical Center. The study protocol adhered to the tenets of the Declaration of Helsinki and written informed consent was obtained from all eligible patients.

All the patients underwent a comprehensive ophthalmic examination including visual acuity, refractive error measurement, and OCT measurement, and the diagnosis of CSC was made based on OCT, fluorescein angiography, and indocyanine green angiography (ICGA) findings. In the current study, 4 patients did not undergo ICGA due to the allergy. For the remaining 71 eyes, we investigated the incidence of CVH. Spectral domain OCT (Spectralis, Heidelberg Engineering) was used to measure the FRT, and the SFCT was estimated using an EDI technique. EDI-OCT examinations were performed between 9:00 and 11:00 a.m. because choroidal thickness is known to exhibit diurnal fluctuations [14]. The exclusion criteria were as follows: (1) the absence of a detailed medical history; (2) a history of previous ocular surgery (other than uncomplicated cataract surgery) or other retinal disorders; (3) the presence of high myopia (−6.0 diopter or greater), and (4) the presence of choroidal neovascularization or polypoidal choroidal vasculopathy, and (5) a history of steroid use.

Binarization of the OCT images was performed to calculate the total choroidal area (TCA), stromal area (SA), luminal area (LA), and CVI, as previously reported [15]. Briefly, the

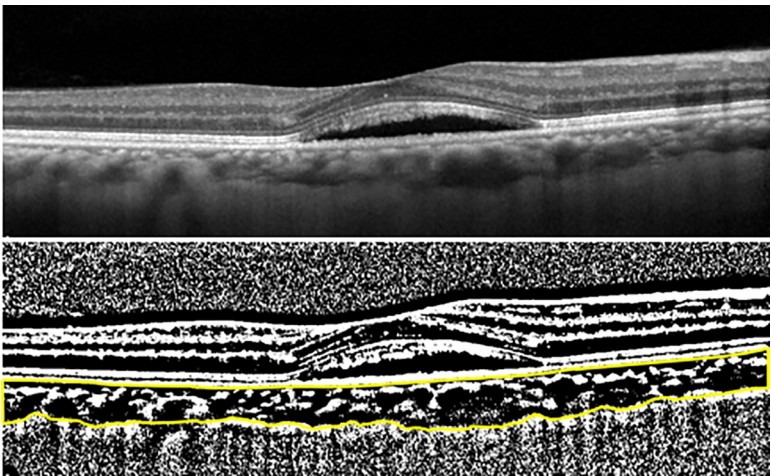

**Fig 1. Binarization of an OCT image in a patient with CSC.** (A) Representative image of the horizontal EDI-OCT scan in CSC eye with serous retinal detachment. (B) Binarized OCT image using Niblack method. Yellow line indicates the border of the choroid. Subsequently, the CVI was calculated as luminal (black) pixels/total choroidal pixels in each eye. CSC, central serous chorioretinopathy; CVI, choroidal vascular index.

choroidal area of the horizontal EDI-OCT images across the fovea was binarized using the Niblack method with ImageJ software (**Fig 1**). The images were converted to 8 bits, and the Niblack auto-local threshold was applied to binarize the images to separate the choroidal luminal and stromal areas. Then, CVI was calculated as luminal pixels/total choroidal pixels in total choroidal area with a width of 6,000μm.

## Statistical analyses

To investigate the relationship between smoking and CSC types (classic or chronic), the chi-squared test was conducted. Baseline parameters (age, refractive error, logMAR visual acuity, FRT, SFCT and CVI) were compared between smokers and non-smokers using the exact Wilcoxon rank sum test. In addition, the associations between SFCT and CVI and the baseline parameters (age, refractive error, and history of smoking) were analyzed using a multivariate linear regression. Subsequently, model selection was performed to identify the optimal linear regression model using the second-order bias-corrected Akaike's information criterion (AICc) index from all $2^4$ patterns consisting of four variables (age, refractive error, history of smoking/hypertension). The AIC is a well-known statistical measurement used in model selection, and the AICc is a corrected version of the AIC that provides an accurate estimation even when the sample size is small [16,17]. The variables selected using this model were regarded as being statistically significant. All statistical analyses were performed using the statistical programming language R (ver. 3.4.3, The R Foundation for Statistical Computing, Vienna, Austria).

## Results

**Table 1** shows the baseline characteristics of the patients in the present study. Among 75 eyes in 75 CSC patients (58 males and 17 females) with serous retinal detachment (SRD), 19 patients were classic CSC and 56 were chronic CSC. 45 patients were smokers and 30 patients were non-smokers. In the smoking group, 10 eyes were classic CSC and 35 eyes were chronic CSC. On the other hand, 9 eyes were classic and 21 eyes were chronic CSC in non-smoking group. There was no significant relationship between smoking and CSC type (p = 0.63, chi-squared test). Thirty-one patients were current smokers, and the remaining 14 patients were

**Table 1. Baseline characteristics of patients.**

| Parameter | Smoking group | Non-smoking group | P value |
|---|---|---|---|
| Number of patients, eyes | 45, 45 | 30, 30 | - |
| Age (years) | 47.2 ± 6.3 | 46.8 ± 5.5 | 0.74 |
| BI | 221.9 ± 238.6 | – | - |
| Refraction (diopter) | -1.31 ± 1.80 | -2.06 ± 1.84 | 0.10 |
| LogMAR VA | 0.057 ± 0.24 | -0.000089 ± 0.21 | 0.11 |
| SFCT (µm) | 382.0 ± 68.2 | 339.3 ± 52.3 | 0.0038 |
| CVI (%) | 64.6 ± 2.1 | 65.3 ± 2.3 | 0.32 |
| FRT (µm) | 367.8 ± 106.5 | 360.3 ± 130.1 | 0.69 |

BI: Brinkman index, logMAR VA: Logarithm of the minimum angle of resolution of visual acuity, SFCT: Subfoveal choroidal thickness, FRT: Foveal retinal thickness, CVI: Choroidal vascular index.

non-current smokers. The mean patient age was 47.1 ± 6.0 years (mean ± standard deviation). No significant difference in age was seen between the smoking group (47.2 ± 6.3 years) and the non-smoking group (46.8 ± 5.5 years, $p = 0.74$, Wilcoxon rank sum test). The Brinkman index (BI, number of cigarettes smoked per day multiplied by the number of years of smoking) was 221.9 ± 238.6 in the smokers group. The mean spherical equivalent of the refractive error also showed no significant difference between the smoking and non-smoking groups (-1.31 ± 1.80 diopter vs. -2.06 ± 1.84 diopter, $p = 0.10$). A significant difference in the SFCT was seen between the smoker and non-smoker groups (382.0 ± 68.2 µm vs. 339.3 ± 52.3 µm, $p = 0.0038$, Wilcoxon rank sum test, **Fig 2A**), while no significant difference was observed in the CVI (64.6 ± 2.1% vs. 65.3 ± 2.3%, $p = 0.32$, **Fig 2B**). Forty-one of 45 smoking patients (91.1%) had CVH and 23 of 26 non-smoking patients (88.5%) had CVH. The prevalence of CVH was numerically higher in smoking group than that in non-smoking group, however, the difference was not statistically significant ($p = 0.95$, chi-squared test).

We then investigated factors associated with SFCT. As a result of AICc model selection, the optimal model for SFCT included the variables of age, refractive error expressed by the

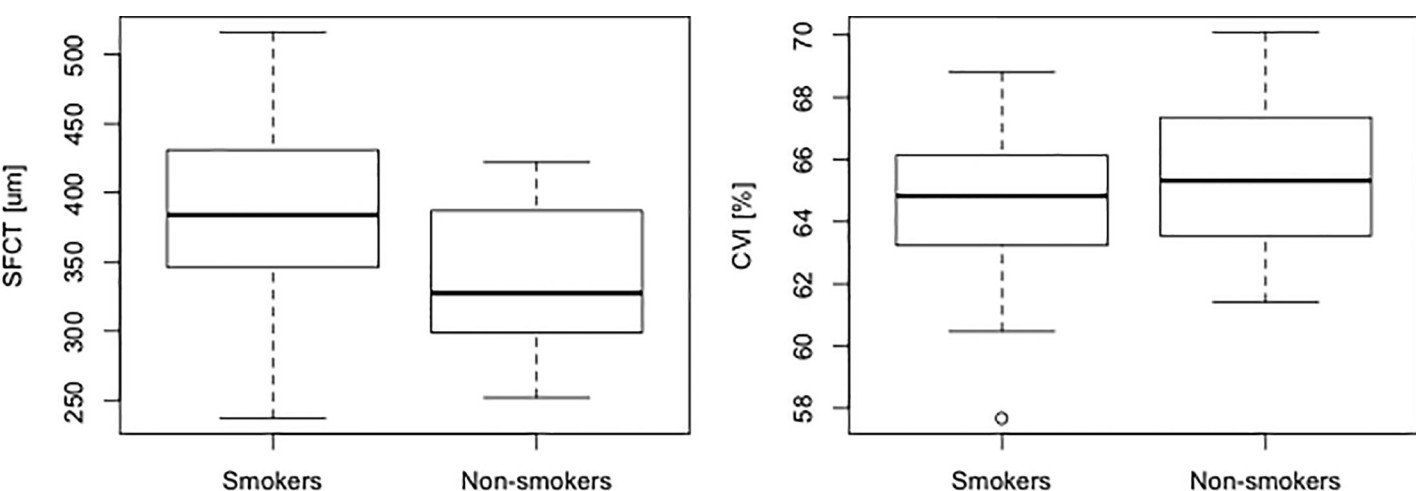

**Fig 2. Comparison of SFCT and CVI between smokers and non-smokers in patients with CSC.** A significant difference in SFCT was seen between smokers and non-smokers (A, $p = 0.0038$, Wilcoxon rank sum test), but no significant difference in CVI was seen (B, $p = 0.32$, Wilcoxon rank sum test). SFCT, subfoveal choroidal thickness; CVI, choroidal vascular index; CSC, central serous chorioretinopathy.

**Table 2. Relationship between smoking and SFCT.**

| Variables | Univariate analysis | | | The optimal model | | |
|---|---|---|---|---|---|---|
| | Coefficient | Stderr | P value | Coefficient | Stderr | P value |
| Age | -3.39 | 1.22 | 0.0070 | -3.40 | 1.12 | 0.0034 |
| SE | 10.64 | 3.97 | 0.0091 | 8.27 | 3.71 | 0.029 |
| History of smoking | 42.7 | 14.7 | 0.0049 | 37.7 | 13.9 | 0.0082 |
| History of hypertension | 8.43 | 22.36 | 0.71 | N.S. | N.S. | N.S. |

SFCT: Subfoveal choroidal thickness, Stderr: Standard error, SE: Spherical equivalent, N.S.: Not selected.

spherical equivalent (SE), and history of smoking among the variables of age, SE, and history of smoking or hypertension (**Table 2**). The optimal model formula was as follows:

SFCT = 515.7–3.40 x Age (standard error [Stderr] = 1.12) + 8.27 x SE (Stderr = 3.71) + 37.7 x Smoking (Stderr = 13.9) (AICc = 827.6).

In the aforementioned analysis, a history of smoking, but not BI, was included in the variable due to colinearity between these two factors. When this analysis was performed using the BI instead of a history of smoking (**Table 3**), BI was selected as a predictive variable, and the formula for the optimal model for SFCT was as follows:

SFCT = 568.5–4.47 x Age (Stderr = 1.19) + 7.79 x SE (Stderr = 3.72) + 0.088 x BI (Stderr = 0.030) (AICc = 826.7).

On the other hand, the optimal model for CVI included SE and history of smoking among the baseline parameters. The formula for the optimal model was as follows:

CVI = 65.9 + 0.29 x SE (Stderr = 0.14)– 0.90 x Smoking (Stderr = 0.51) (AICc = 331.7)

However, unlike the results for SFCT, when the BI was used instead of a history of smoking, BI was not selected as an explanatory variable for CVI.

Moreover, we investigated the relationship between current smoking and the choroidal structure. No significant differences in SFCT and CVI were seen between current smokers and non-current smokers ($p$ = 0.56 [**Fig 3**] and 0.98, respectively, Wilcoxon rank sum test). Furthermore, current smoking was not selected for the SFCT and the CVI using AICc model selection.

## Discussion

In the present study, the relationship between the choroidal structure and cigarette smoking was investigated in patients with CSC. As a result, a history of smoking was significantly correlated with an increased SFCT in CSC eyes with subretinal fluid (SRF). Importantly, the effect of smoking on the SFCT was dose (pack years)—dependent as shown in the prediction model,

**Table 3. Relationship between BI and SFCT.**

| Variables | Univariate analysis | | | The optimal model | | |
|---|---|---|---|---|---|---|
| | Coefficient | Stderr | P value | Coefficient | Stderr | P value |
| Age | -3.39 | 1.22 | 0.0070 | -4.47 | 1.19 | 0.00034 |
| SE | 10.64 | 3.97 | 0.0091 | 7.79 | 3.72 | 0.040 |
| BI | 0.063 | 0.031 | 0.046 | 0.088 | 0.030 | 0.0052 |
| History of hypertension | 8.43 | 22.36 | 0.71 | N.S. | N.S. | N.S. |

BI: Brinkman index, SFCT: Subfoveal choroidal thickness, Stderr: Standard error, SE: Spherical equivalent, N.S.: Not selected.

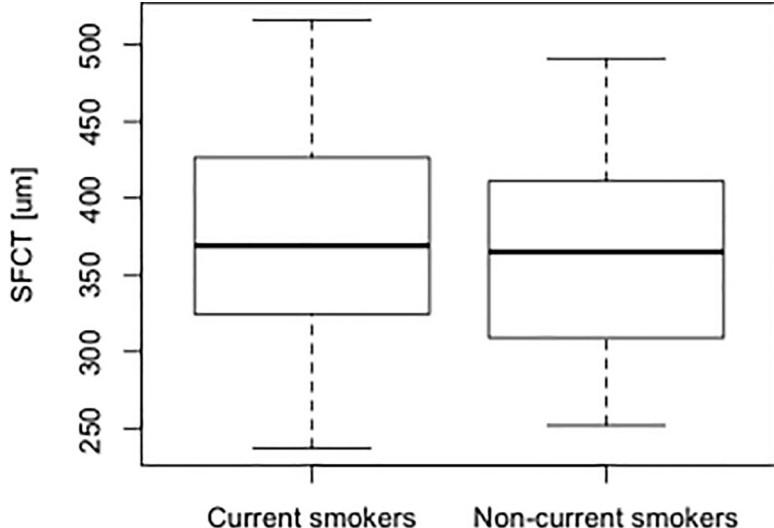

**Fig 3. Comparison of SFCT between current and non-current smokers in patients with CSC.** No significant difference in SFCT was seen between current and non-current smokers ($p$ = 0.56, Wilcoxon rank sum test). SFCT, subfoveal choroidal thickness; CSC, central serous chorioretinopathy.

suggesting the association was not a mere chance finding. On the other hand, the presence of a history of smoking was associated with a decreased CVI.

The subfoveal choroid is significantly thicker in eyes with CSC [18,19], and the resolution of CSC is reportedly correlated with a reduction in the SFCT [20,21]. Moreover, a previous report suggested that the CVI was higher in acute CSC eyes, compared with contralateral non-CSC eyes [7]. Our present results demonstrated that the presence of a history of smoking was correlated with an increased SFCT and a decreased CVI, suggesting that the pathology of CSC may be somewhat different between smokers and non-smokers. Although there is no direct link between visual acuity and the choroidal thickness, corroborating evidence from prior observations [18–21] suggest that choroidal thickness may be associated with disease activity; further longitudinal study would be of interest to examine the effect of smoking on visual outcome in CSC patients. Previous studies have investigated the effect of smoking on the CVI, with varying results in normal subjects. Agrawal et al. reported that a univariate analysis demonstrated a significant correlation between cigarette smoking and the CVI, while a multivariate analysis showed no significant correlation [22]. On the contrary, Wei et al. suggested that smoking was associated with a smaller CVI in normal eyes [12]. Regarding CVI in CSC patients, a previous study reported that the CVI was increased in patients with acute CSC, compared with contralateral non-CSC eyes, suggesting the possibility that the CVI was influenced by the activity of CSC and vice versa [7]. In contrast, our present results for eyes with CSC suggested that the CVI was negatively associated with a smoking history. Recently, CSC has been considered as one of the pachychoroid spectrum diseases with focal or diffuse increase in choroidal thickness, dilated large choroidal vessels, and CVH. In pachychoroid spectrum diseases including CSC and polypoidal choroidal vasculopathy, CVH are associated with an increased SFCT and an increased CVI [23–25]. In the current study, the prevalence of CVH was compared between the smoking and non-smoking groups in eyes with CSC. As a result, the prevalence of CVH was higher in smoking group than in non-smoking group, however this difference was not significant. This may be because the frequency of CVH is high in eyes with CSC in general, which masked the difference between the smoking and non-

smoking groups. It should be noted that the reason why a smoking habit causes the decreased CVI still remains unclear, and further studies are needed shedding light on this issue.

The present study had some limitations. First, this study was retrospective in nature, and the number of subjects was relatively small. Second, the choroidal structure in eyes with CSC is associated with the disease duration [26], however, this information was not available in the current study. Smoking cessation is usually recommended to CSC patients in clinical settings, however its effectiveness remains elusive. It would be interesting to examine the choroidal structural changes in eyes with CSC that are caused by smoking cessation.

In conclusion, cigarette smoking was associated with an increased subfoveal choroidal thickness in CSC patients. Smoking might be an important factor to modulate CSC by influencing the choroidal thickness.

## Supporting information

**S1 File.**
(CSV)

## Author Contributions

**Conceptualization:** Tatsuya Inoue, Ryo Asaoka, Ryo Obata, Maiko Maruyama-Inoue, Yasuo Yanagi, Kazuaki Kadonosono.

**Data curation:** Rei Arasaki, Shouko Ikeda, Arisa Ito.

**Investigation:** Kazuyoshi Okawa, Tatsuya Inoue, Ryo Asaoka, Arisa Ito.

**Methodology:** Tatsuya Inoue, Keiko Azuma.

**Visualization:** Rei Arasaki, Arisa Ito.

**Writing – original draft:** Kazuyoshi Okawa, Tatsuya Inoue, Rei Arasaki, Shouko Ikeda, Arisa Ito.

**Writing – review & editing:** Tatsuya Inoue, Ryo Asaoka, Ryo Obata, Maiko Maruyama-Inoue, Yasuo Yanagi, Kazuaki Kadonosono.

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
