## [Decision Letter · Decision Letter 0]

22 Jan 2021

PONE-D-20-39783

Correlation between choroidal structure and smoking in eyes with central serous chorioretinopathy

PLOS ONE

Dear Dr. Inoue,

Thank you for submitting your manuscript to PLOS ONE. After careful consideration, we feel that it has merit but does not fully meet PLOS ONE’s publication criteria as it currently stands. Therefore, we invite you to submit a revised version of the manuscript that addresses the points raised during the review process.

this is an important study that both reviewers found worthwhile but made several comments to improve it 

We  look forward to the revised version

We look forward to receiving your revised manuscript.

Kind regards,

Demetrios G. Vavvas

Academic Editor

PLOS ONE

Journal Requirements:

Reviewers' comments:

Reviewer's Responses to Questions

**Comments to the Author**

1. Is the manuscript technically sound, and do the data support the conclusions?

Reviewer #1: Partly

Reviewer #2: Yes

2. Has the statistical analysis been performed appropriately and rigorously? 

Reviewer #1: Yes

Reviewer #2: Yes

3. Have the authors made all data underlying the findings in their manuscript fully available?

Reviewer #1: Yes

Reviewer #2: Yes

4. Is the manuscript presented in an intelligible fashion and written in standard English?

Reviewer #1: Yes

Reviewer #2: Yes

5. Review Comments to the Author

Reviewer #1: Many thanks for the opportunity to review this interesting manuscript entitled “Correlation between choroidal structure and smoking in eyes with central serous chorioretinopathy”. This study investigated the effect of smoking on the choroidal structure in patients with CSC. The authors concluded that cigarette smoking was associated with an increased choroidal thickness in patients with CSC. Although this manuscript is interesting, I take this opportunity to comment on some issues.

1. Line 162: Although this study included 75 eyes with CSC, the authors did not describe the detail of the subjects. How many eyes with classic CSC or chronic CSC were included in smoking and non-smoking groups? How many patients had used steroid? Those characteristics reportedly affect the choroidal thickness and choroidal vascular index in eyes with CSC, major results in this study.

2. Line 165: “Thirty-one patients were current smokers, and the remaining 44 patients were non-current smokers.” Should be “Thirty-one patients were current smokers, and the remaining 14 patients were non-current smokers.”

3. Line 124: The exclusion criteria included the presence of high myopia (−6.0 diopter or greater). Were the eyes analyzed in this study all phakic? If pseudo-phakic eyes were included, how were the eyes with high myopia excluded? Did the authors measure the axial length?

4. Line 126: “the presence of choroidal neovascularization and polypoidal choroidal vasculopathy.” should be “the presence of choroidal neovascularization or polypoidal choroidal vasculopathy.”

5. Line 130: The authors should clarify the range of horizontal EDI-OCT image analyzed to calculate the choroidal vascular index.

6. Line 222: The authors stated “the presence of a history of smoking was associated with a decreased CVI”. However, the formula for the optimal model was CVI=65.9+0.29 x SE (Stderr=0.14, p=0.039) –0.90 x Smoking (Stderr=0.51, p=0.083). This formula means the history of smoking was not a significant factor for CVI.

7. Line 233: There are no evidences supporting the following statement: our findings may in part explain why the smoking was reportedly associated with poor visual outcome in CSC patients.

8. Line 244: The authors speculated that choroidal vascular hyperpermeability (CVH) might result in the lower CVI in patients with smoking history. In this study, all patients underwent indocyanine green angiography (ICGA). Therefore, the incidence of CVH in ICGA should be assessed. If the hypothesis is correct, the incidence of CVH should be significantly higher in smoking group compared to non-smoking group. In my opinion, if the fluid is accumulated in the choroid secondary to CVH, CVI might increase. Furthermore, previous studies investigating CSC eyes reported that the eyes with CVH showed significantly greater choroidal thickness compared to the eyes without CVH.

Reviewer #2: Dear Authors,

I have read your manuscript entitled “Correlation between choroidal structure and smoking in eyes with central serous chorioretinopathy”. The manuscript is written in a concise and clear manner and the conclusions are well supported by the presented data. Besides a couple of comments below, I have no major concerns and feel this manuscript fulfills the criteria for publication in PLOS ONE.

The only minor comment

Page 10 table 1 legend - please include the BI definition

Page 10 lines 165-166 - while previously stated 45 subjects were smokers and 30 - non-smoker, it is unclear what the following means: “31 patients were current smokers and 44 were non-current..”; the sum gives us 75 patients, does it mean there all smokers in the past, are these the same patients or different

6. PLOS authors have the option to publish the peer review history of their article (what does this mean?). If published, this will include your full peer review and any attached files.

Reviewer #1: No

Reviewer #2: No

---

## [Author Response · Author response to Decision Letter 0]

29 Jan 2021

Dear Prof. Vavvas

We appreciate you and the reviewers for the constructive and insightful comments. Please find our enclosed point-by-point responses (in red) to the reviewers’ comments. We hope that this revision will meet your expectations, and that our revised manuscript will better suit for publication on PLOS ONE.

Thank you for submitting your manuscript to PLOS ONE. After careful consideration, we feel that it has merit but does not fully meet PLOS ONE’s publication criteria as it currently stands. Therefore, we invite you to submit a revised version of the manuscript that addresses the points raised during the review process.

this is an important study that both reviewers found worthwhile but made several comments to improve it. We look forward to the revised version

Thank you very for your insightful comments. Please see our responses below.

Reviewer #1: Many thanks for the opportunity to review this interesting manuscript entitled “Correlation between choroidal structure and smoking in eyes with central serous chorioretinopathy”. This study investigated the effect of smoking on the choroidal structure in patients with CSC. The authors concluded that cigarette smoking was associated with an increased choroidal thickness in patients with CSC. Although this manuscript is interesting, I take this opportunity to comment on some issues.

Thank you very for your constructive and insightful comments. Please see our responses to each comment.

1. Line 162: Although this study included 75 eyes with CSC, the authors did not describe the detail of the subjects. How many eyes with classic CSC or chronic CSC were included in smoking and non-smoking groups? How many patients had used steroid? Those characteristics reportedly affect the choroidal thickness and choroidal vascular index in eyes with CSC, major results in this study.

Thank you for very much your insightful comment. 19 classic CSC and 56 chronic CSC were enrolled in the current study. And patients who had used steroid were excluded from the current study. We have changed our manuscript as follows:

“Among 75 eyes in 75 CSC patients (58 males and 17 females) with serous retinal detachment (SRD), 19 patients were classic CSC and 56 were chronic CSC.”

“The exclusion criteria were as follows: (1) the absence of a detailed medical history; (2) a history of previous ocular surgery (other than uncomplicated cataract surgery) or other retinal disorders; (3) the presence of high myopia (−6.0 diopter or greater), and (4) the presence of choroidal neovascularization or polypoidal choroidal vasculopathy, and (5) a history of steroid use.”

2. Line 165: “Thirty-one patients were current smokers, and the remaining 44 patients were non-current smokers.” Should be “Thirty-one patients were current smokers, and the remaining 14 patients were non-current smokers.”

Thank you for your suggestion. We have revised our manuscript.

3. Line 124: The exclusion criteria included the presence of high myopia (−6.0 diopter or greater). Were the eyes analyzed in this study all phakic? If pseudo-phakic eyes were included, how were the eyes with high myopia excluded? Did the authors measure the axial length?

Thank you for the critical comment. The current study included 74 phakic eyes and only 1 pseudophakic eye. About the pseudophakic eyes, we measured axial length and denied the possibility that they were high myopic eyes. 

4. Line 126: “the presence of choroidal neovascularization and polypoidal choroidal vasculopathy.” should be “the presence of choroidal neovascularization or polypoidal choroidal vasculopathy.”

We have changed the manuscript according to your comment. Thank you very much.

5. Line 130: The authors should clarify the range of horizontal EDI-OCT image analyzed to calculate the choroidal vascular index.

Thank you for the comment. The range of analyzed region was 6000μm wide. We have edited the manuscript in the Methods section according to the suggestion. 

“Then, CVI was calculated as luminal pixels/total choroidal pixels in total choroidal area with a width of 6,000μm.” 

6. Line 222: The authors stated “the presence of a history of smoking was associated with a decreased CVI”. However, the formula for the optimal model was CVI=65.9+0.29 x SE (Stderr=0.14, p=0.039) –0.90 x Smoking (Stderr=0.51, p=0.083). This formula means the history of smoking was not a significant factor for CVI.

Thank you for your insightful suggestion. We apologize for the confusion. The optimal model was calculated by AICc model selection as mentioned in Methods section. Thus, we did not use p value in the estimation of the effect of variables, regardless of the 'by-product' of p values in the current analysis (Hasley, L. G. (2019). The reign of the p-value is over: what alternative analyses could we employ to fill the power vacuum? Biology Letters, 15. doi 10.1098/rsbl.2019.0174). To avoid confusion, we removed the p value descriptions.

“ CVI=65.9+0.29 x SE (Stderr=0.14) –0.90 x Smoking (Stderr=0.51) ” 

7. Line 233: There are no evidences supporting the following statement: our findings may in part explain why the smoking was reportedly associated with poor visual outcome in CSC patients.

We appreciate the reviewer for raising this point. We agree the reviewer’s comment and have revised our manuscript as follows;

“further longitudinal study would be of interest to examine the effect of smoking on visual outcome in CSC patients.” 

8. Line 244: The authors speculated that choroidal vascular hyperpermeability (CVH) might result in the lower CVI in patients with smoking history. In this study, all patients underwent indocyanine green angiography (ICGA). Therefore, the incidence of CVH in ICGA should be assessed. If the hypothesis is correct, the incidence of CVH should be significantly higher in smoking group compared to non-smoking group. In my opinion, if the fluid is accumulated in the choroid secondary to CVH, CVI might increase. Furthermore, previous studies investigating CSC eyes reported that the eyes with CVH showed significantly greater choroidal thickness compared to the eyes without CVH.

Thank you for the insightful and constructive suggestion. As you suggested, we investigated the incidence of CVH on ICGA. In the current study, 4 patients did not undergo ICGA due to the allergy. For the remaining 71 eyes, we investigated the incidence of CVH according to the reviewer’s recommendation. As a consequence, 41 of 45 smoking patients (91.1%) had CVH and 23 of 26 non-smoking patients (88.5%) had CVH. The prevalence of CVH was numerically higher in smoking group than that in non-smoking group, however, the difference was not significant presumably due to the fact that the frequency of CVH is basically high in eyes with CSC, even in those who never smoked, and therefore any additional effects would be difficult to assess due to the ceiling effect. We understand your hypothesis that if the fluid is accumulated in the choroid secondary to CVH, CVI might increase. However, due to the inherent difficulties in assessing the CVH in CSC on ICGA, to the best of our knowledge, there was no report to investigate the correlation between CVH and CVI in eyes with CSC. Additionally, it is possible that there is relative decrease in CVI due to the exudative change happening in the choroidal stroma in CSC. In support of this, in some inflammatory diseases, there is an increased choroidal thickness with concomitant decrease in CVI (Agarwal A et al. Choroidal structural changes in tubercular multifocal serpinginoid choroiditis. 2018 Ocul Immunol Inflamm. 25;134-145.). Further investigations are needed to clarify the relationship between CVH and CVI in CSC patients, however, the present study did not enroll normal control subjects so this is beyond the scope of our present study. Again, thank you very much for your insightful suggestion. We toned down our strong sentences according to your comments.

Reviewer #2: Dear Authors,

I have read your manuscript entitled “Correlation between choroidal structure and smoking in eyes with central serous chorioretinopathy”. The manuscript is written in a concise and clear manner and the conclusions are well supported by the presented data. Besides a couple of comments below, I have no major concerns and feel this manuscript fulfills the criteria for publication in PLOS ONE.

The only minor comment

Page 10 table 1 legend - please include the BI definition

Thank you for your comment. We have revised our manuscript according to the reviewer’s comment.

Page 10 lines 165-166 - while previously stated 45 subjects were smokers and 30 - non-smoker, it is unclear what the following means: “31 patients were current smokers and 44 were non-current.”; the sum gives us 75 patients, does it mean there all smokers in the past, are these the same patients or different

Thank you very much. These are the same patients and it means that 31 of 45 CSC patients were current smokers and remaining 14 patients were non-current past smokers. Therefore, we have revised the manuscript in Results section.

---

## [Decision Letter · Decision Letter 1]

15 Feb 2021

PONE-D-20-39783R1

Correlation between choroidal structure and smoking in eyes with central serous chorioretinopathy

PLOS ONE

Dear Dr. Inoue,

Thank you for submitting your manuscript to PLOS ONE. After careful consideration, we feel that it has merit but does not fully meet PLOS ONE’s publication criteria as it currently stands. Therefore, we invite you to submit a revised version of the manuscript that addresses the points raised during the review process.

The first reviewer has some more questions that require to be addressed before a final decision can be made. Can you please address these comments? Thank you 

We look forward to receiving your revised manuscript.

Kind regards,

Demetrios G. Vavvas

Academic Editor

PLOS ONE

Reviewers' comments:

Reviewer's Responses to Questions

**Comments to the Author**

1. If the authors have adequately addressed your comments raised in a previous round of review and you feel that this manuscript is now acceptable for publication, you may indicate that here to bypass the “Comments to the Author” section, enter your conflict of interest statement in the “Confidential to Editor” section, and submit your "Accept" recommendation.

Reviewer #1: (No Response)

Reviewer #2: All comments have been addressed

2. Is the manuscript technically sound, and do the data support the conclusions?

Reviewer #1: Partly

Reviewer #2: Yes

3. Has the statistical analysis been performed appropriately and rigorously? 

Reviewer #1: I Don't Know

Reviewer #2: I Don't Know

4. Have the authors made all data underlying the findings in their manuscript fully available?

Reviewer #1: No

Reviewer #2: Yes

5. Is the manuscript presented in an intelligible fashion and written in standard English?

Reviewer #1: Yes

Reviewer #2: Yes

6. Review Comments to the Author

Reviewer #1: 1. It has been reported that the choroidal thickness and choroidal vascular index are significantly different between classic and chronic CSC. However, the authors did not state the distribution of the CSC types in smoking and non-smoking groups. The distribution might affect the main results in this study.

2. There are no evidences supporting the following statement in the discussion: luminal area increase in the smoking group was accompanied by an increase in the stromal area as well because of the increased choroidal vascular permeability in CSC patients. Why was the luminal area increased in the smoking group? Moreover, in the response letter, the authors mentioned that increased choroidal thickness with concomitant decrease in CVI was reportedly observed in some inflammatory diseases, which might support their hypothesis. However, the pathophysiology in such inflammatory diseases is quite different from that in CSC.

Reviewer #2: Dear Editor,

Thank you for this opportunity to review the revised manuscript entitled “Correlation between choroidal structure and smoking in eyes with central serous chorioretinopathy”.

All the comments are properly addressed. While responding to a very good point from reviewer 1 authors investigated the incidence of CVH as assessed by ICGA. Would suggest including it in the manuscript ( main or supplement).

7. PLOS authors have the option to publish the peer review history of their article (what does this mean?). If published, this will include your full peer review and any attached files.

Reviewer #1: No

Reviewer #2: No

---

## [Author Response · Author response to Decision Letter 1]

17 Feb 2021

Dear Prof. Demetrios G. Vavvas

We appreciate you and the reviewers for the constructive and insightful comments. Please find our enclosed point-by-point responses to the reviewers’ comments. We hope that this revision will meet your expectations, and that our revised manuscript will better suit for publication on PLOS ONE.

Reviewer #1: 1. It has been reported that the choroidal thickness and choroidal vascular index are significantly different between classic and chronic CSC. However, the authors did not state the distribution of the CSC types in smoking and non-smoking groups. The distribution might affect the main results in this study.

Thank you for the critical and constructive suggestion. In the smoking group, 10 eyes were classic CSC and 35 eyes were chronic CSC. In non-smoking group, 9 eyes were classic and 21 eyes were chronic CSC. To analyze the relationship between smoking status and CSC type (classic or chronic), the chi-squared test was conducted. As a result, there was no significant relationship between smoking and CSC type (p=0.63), which would imply that the distribution of the CSC type has only a negligible effect on the current results. We have revised our manuscript in the Results section.

2. There are no evidences supporting the following statement in the discussion: luminal area increase in the smoking group was accompanied by an increase in the stromal area as well because of the increased choroidal vascular permeability in CSC patients. Why was the luminal area increased in the smoking group? Moreover, in the response letter, the authors mentioned that increased choroidal thickness with concomitant decrease in CVI was reportedly observed in some inflammatory diseases, which might support their hypothesis. However, the pathophysiology in such inflammatory diseases is quite different from that in CSC.

Thank you for the insightful comment. We added this speculation, because we speculated that the luminal area increase in the smoking group was accompanied by an increase in the stromal area due to the increased choroidal vascular permeability, basing on a previous report by Wei et al. in which it was suggested that the increased stromal area in smoking group may be attributed to a chronic proinflammatory response (Ref 12). Nonetheless, we now totally agree with the reviewer’s comment. Furthermore, in the current study, we did not clarify this hypothesis. Moreover, we also agree with the reviewer’s comment that the pathophysiology of CSC is different from that of inflammatory disease. Thus, the too much speculative description in the Discussion is now deleted in the revised manuscript.

Reviewer #2: Dear Editor,

Thank you for this opportunity to review the revised manuscript entitled “Correlation between choroidal structure and smoking in eyes with central serous chorioretinopathy”.

All the comments are properly addressed. While responding to a very good point from reviewer 1 authors investigated the incidence of CVH as assessed by ICGA. Would suggest including it in the manuscript ( main or supplement).

Thank you for the insightful comment. We have added the following sentences, according to the reviewer’s recommendation. 

Results

“Forty-one of 45 smoking patients (91.1%) had CVH and 23 of 26 non-smoking patients (88.5%) had CVH. The prevalence of CVH was numerically higher in smoking group than that in non-smoking group, however, the difference was not statistically significant.”

Discussion

“Recently, CSC has been considered as one of the pachychoroid spectrum diseases with focal or diffuse increase in choroidal thickness, dilated large choroidal vessels, and CVH. In pachychoroid spectrum diseases including CSC and polypoidal choroidal vasculopathy, CVH are associated with an increased SFCT and an increased CVI (23-25). In the current study, the prevalence of CVH was compared between the smoking and non-smoking groups in eyes with CSC. As a result, the prevalence of CVH was higher in smoking group than in non-smoking group, however this difference was not significant. This may be because the frequency of CVH is high in eyes with CSC in general, which masked the difference between the smoking and non-smoking groups. It should be noted that the reason why a smoking habit causes the decreased CVI still remains unclear, and further studies are needed shedding light on this issue.”

---

## [Decision Letter · Decision Letter 2]

11 Mar 2021

Correlation between choroidal structure and smoking in eyes with central serous chorioretinopathy

PONE-D-20-39783R2

Dear Dr. Inoue,

We’re pleased to inform you that your manuscript has been judged scientifically suitable for publication and will be formally accepted for publication once it meets all outstanding technical requirements.

Kind regards,

Demetrios G. Vavvas

Academic Editor

PLOS ONE

Additional Editor Comments (optional):

Reviewers' comments:

Reviewer's Responses to Questions

**Comments to the Author**

1. If the authors have adequately addressed your comments raised in a previous round of review and you feel that this manuscript is now acceptable for publication, you may indicate that here to bypass the “Comments to the Author” section, enter your conflict of interest statement in the “Confidential to Editor” section, and submit your "Accept" recommendation.

Reviewer #1: All comments have been addressed

Reviewer #2: All comments have been addressed

2. Is the manuscript technically sound, and do the data support the conclusions?

Reviewer #1: Yes

Reviewer #2: Yes

3. Has the statistical analysis been performed appropriately and rigorously? 

Reviewer #1: I Don't Know

Reviewer #2: I Don't Know

4. Have the authors made all data underlying the findings in their manuscript fully available?

Reviewer #1: Yes

Reviewer #2: Yes

5. Is the manuscript presented in an intelligible fashion and written in standard English?

Reviewer #1: Yes

Reviewer #2: Yes

6. Review Comments to the Author

Reviewer #1: The authors have adequately addressed all the comments raised in this round of review. This reviewer feels the revised manuscript is now acceptable for publication.

Reviewer #2: Dear Authors,

i have read your revised manuscript entitled "Correlation between choroidal structure and smoking in eyes with central serous chorioretinopathy"and thank you for responding to all comments.

7. PLOS authors have the option to publish the peer review history of their article (what does this mean?). If published, this will include your full peer review and any attached files.

Reviewer #1: No

Reviewer #2: No

---

## [Editor Report · Acceptance letter]

15 Mar 2021

PONE-D-20-39783R2 

Correlation between choroidal structure and smoking in eyes with central serous chorioretinopathy 

Dear Dr. Inoue:

I'm pleased to inform you that your manuscript has been deemed suitable for publication in PLOS ONE. Congratulations! Your manuscript is now with our production department. 

Kind regards, 

on behalf of

Dr. Demetrios G. Vavvas 

Academic Editor

PLOS ONE